# V2T-GAN: Three-Level Refined Light-Weight GAN with Cascaded Guidance for Visible-to-Thermal Translation

**DOI:** 10.3390/s22062119

**Published:** 2022-03-09

**Authors:** Ruiming Jia, Xin Chen, Tong Li, Jiali Cui

**Affiliations:** School of Information Science and Technology, North China University of Technology, Beijing 100144, China; jiaruiming@ncut.edu.cn (R.J.); xinchen@mail.ncut.edu.cn (X.C.); litong300@pingan.com.cn (T.L.)

**Keywords:** image domain translation, infrared image simulation, generative adversarial network

## Abstract

Infrared image simulation is challenging because it is complex to model. To estimate the corresponding infrared image directly from the visible light image, we propose a three-level refined light-weight generative adversarial network with cascaded guidance (V2T-GAN), which can improve the accuracy of the infrared simulation image. V2T-GAN is guided by cascading auxiliary tasks and auxiliary information: the first-level adversarial network uses semantic segmentation as an auxiliary task, focusing on the structural information of the infrared image; the second-level adversarial network uses the grayscale inverted visible image as the auxiliary task to supplement the texture details of the infrared image; the third-level network obtains a sharp and accurate edge by adding auxiliary information of the edge image and a displacement network. Experiments on the public dataset Multispectral Pedestrian Dataset demonstrate that the structure and texture features of the infrared simulation image obtained by V2T-GAN are correct, and outperform the state-of-the-art methods in objective metrics and subjective visualization effects.

## 1. Introduction

Infrared images are widely used in military, medical, industrial and agricultural fields, and are generally obtained by shooting target scenes with an infrared thermal imager. However, in some special environments, the amount of image data that can be obtained by an infrared thermal imager is relatively insufficient, and the equipment is expensive. These problems limit the acquisition of infrared image data. Therefore, related research on infrared image simulation has been progressing.

The traditional infrared simulation approach can be divided into two types: infrared image simulation based on three-dimensional modeling and infrared image simulation based on visible light image. The first method uses three-dimensional modeling of the scene, and then simulates according to infrared radiation characteristics [1,2,3], without the need for real visible light images. The disadvantage is that the overall process is complicated, and the texture of the simulation result is unnatural. Furthermore, because it only targets a single scene, the generalization performance of the model is poor. The second method requires real visible light images, which is a simpler and more convenient method than the previous one, but it also has the disadvantages of low simulation accuracy and poor generalization ability. In view of the above problems, we aim to provide a more convenient, accurate and robust infrared simulation approach.

Infrared image simulation based on visible light image is a pixel-level image conversion task, which can predict and simulate corresponding infrared images through visible light images. Recently, the pixel-level image conversion task based on the deep learning method has achieved great success, and the algorithm is relatively simple and convenient. Common pixel-level image conversion tasks include monocular depth estimation [4,5], semantic segmentation [6,7], optical flow estimation [8], image style conversion [9], etc. Among them, the first three tasks require high accuracy, and are generally implemented by convolutional neural networks (CNN). Due to the constraint of the objective function, although the methods based on CNN can obtain better results in the objective metrics, the predicted result map generally has the problem of blurred edges and texture loss. In order to solve this problem, some studies have used conditional generative adversarial network (cGAN) to achieve such tasks [9,10]. The generated image of the cGAN have a natural image texture, clear edges, and better visualization. Based on these observations, we use cGAN to achieve the conversion of a visible light image to the corresponding infrared image.

The visible light image and the infrared image have similar structural information, semantic information and edge information, and the grayscale inverted visible (GIV) im-age have a high degree of similarity with the infrared image in visualization and texture details. The GAN designed in this paper uses semantic segmentation images and GIV images as auxiliary tasks, and visible light edge images as auxiliary information, which can realize the conversion of a visible light image to infrared image end-to-end. To improve the efficiency of the algorithm, a variety of light-weight convolutions are used to reduce the amount of overall network parameters.

The contributions of this paper consist of three aspects. First, a three-level refined light-weight GAN with cascaded guidance (V2T-GAN) is proposed. It aims at converting a visible light image to infrared image end-to-end. Second, a three-level network framework for cascading guidance through auxiliary tasks and auxiliary information is proposed. The first-level network of V2T-GAN uses a semantic segmentation image as an auxiliary task to generate a coarse infrared image with a relatively correct structure; the second-level network uses the GIV image as an auxiliary task to supplement the detailed texture information of the infrared image; and the third-level network adds auxiliary information of the edge image and displacement field to obtain a clear and accurate edge. Third, extensive experiments were carried out on the public dataset Multispectral Pedestrian Dataset (MPD) [11], which clearly demonstrate the effectiveness of the proposed V2T-GAN.

The rest of this study is organized as follows. In Section 2, we discuss the related work on infrared image simulation. In Section 3, we introduce the proposed method in detail. In Section 4, we present the main experiments. In Section 5, we conclude with a brief summary and mention future work.

## 2. Related Work

### 2.1. Infrared Image Simulation

The traditional method of infrared simulation based on visible light image is generally divided into two stages: in the first stage, the visible light image needs to be segmented; and in the second stage, the gray-scale mapping relationship between the visible light image and infrared of different objects is established, and then the infrared image is simulated from the segmented visible light image. Zhou et al. [12] used the threshold method to segment the image, and combined the reflectivity of the ground object to establish the gray-scale mapping relationship, so as to obtain the simulated image. Li et al. [13] achieved image segmentation through a pulse-coupled neural network, after artificially calibrating the material to obtain simulation results through radiation calculation. Infrared imaging systems and visible light imaging systems are both complex and affected by multiple variables. This feature makes it difficult to express the mapping relationship between visible light images and infrared images with a unified formula. Therefore, the model generalization ability of this type of method is poor, and the simulation result lacks natural image texture information.

### 2.2. Pixel-Level Image Conversion Tasks

Image conversion tasks include image style conversion, image perspective conversion, depth estimation, optical flow estimation, semantic segmentation and so on. Generating a corresponding infrared image from a visible light image can be regarded as a mapping from the visible light image domain to the infrared image domain, which is a kind of image domain conversion task. In recent years, deep learning methods have achieved good research results in image domain conversion tasks, such as monocular depth estimation and image style conversion. In [4], Eigen et al. first used an end-to-end CNN to predict the depth map. In further works, some people introduced concepts such as an attention mechanism and continuous conditional random field to improve the performance of the algorithm [14,15,16], whereas some achieved better results by optimizing the network structure [5]. In addition, there is also the use of multi-task [8,17,18,19] learning methods to obtain auxiliary information.

### 2.3. Conditional Generative Adversarial Network

Unlike the image depth estimation task, the conversion from visible light to infrared requires better objective evaluation results, as well as better visual effects. Constrained by the objective function, although the CNN method in the image depth estimation task has obtained good results in objective evaluation indicators, the output image is relatively blurry and loses many texture details. In the field of deep learning, the cGAN derived from the GAN [20] has excellent performance in the image domain conversion task [9,20,21,22,23,24], and the visual effect of the output image is better.

### 2.4. Lightweight Network

Various computer vision tasks implemented through deep learning have shortcomings, such as high network model redundancy and complex calculations. With the development of deep learning, the lightweight and high-efficiency of the network models has become more and more important. Currently, lightweight and efficient network structures are mostly used in tasks such as image classification, object detection and semantic segmentation. Moreover, most lightweight network models use lightweight convolution instead of standard convolution to build network structures. Zhang et al. [25] proposed ShuffleNet with efficient computing power to realize image classification. In [26], Krizhevsky et al. proposed group convolution (GConv) and constructed a network model that includes group convolution, which has fewer network model parameters and higher accuracy than ShuffleNet. MobileNetV2 [27] included Depthwise Separable Convolution (DSConv), which is a lightweight network model that can be applied to mobile architectures. Mehta et al. [28] proposed depthwise dilated separable convolution (DDSConv) and used it to construct the network ESPNetv2, which has high computational efficiency and has a large receptive field. Haase et al. [29] analyzed the depth separable convolution and improved it to obtain the blueprint separable convolution (BSConv). Han et al. [30] proposed the Ghost module, which can effectively reduce the redundancy of feature maps through simple linear operations, thereby greatly improving network computing efficiency.

## 3. V2T-GAN

In order to achieve the conversion from a visible light image to infrared image, we propose a three-level refined light-weight GAN with cascaded guidance. As shown in Figure 1, V2T-GAN is a three-level cascaded network. The first-level network uses semantic segmentation images as an auxiliary task to guide G_1_ to learn infrared images with more accurate structural information; the second-level network uses GIV images as an auxiliary task to guide G_2_ to learn more accurate infrared images with detailed textures; and the third-level network uses visible light edge images as auxiliary information to further optimize the predicted infrared images. G*_d_* predicts the displacement offset map of the second-level network’s output image T_2_ in the *x* and *y* directions, and then resamples T_2_ according to the displacement offset information to obtain the final infrared image T_3_.

### 3.1. First-Level Network

The first-level network uses semantic segmentation images as auxiliary tasks to guide the first-level target task generative network to predict infrared images with more correct structure information. As shown in Figure 1, the blue part is the first-level network, including a target task generator G_1_, a discriminator D_1_ and an auxiliary task network G*_s_*. G_1_ estimates the corresponding infrared image T_1_ from the visible light image, and D_1_ is responsible for identifying the authenticity of the predicted infrared image T_1_ and the target infrared image T*_true_*. Then, G*_s_* estimates the semantic segmentation image from T_1_, and guides G_1_ to pay more attention to the structure information by predicting the semantic segmentation image, thereby predicting the infrared image with more correct structure information.

The network structure of G_1_ and G*_s_* is the generator U-net [31] in pix2pix [9], and the network structure of D_1_ is the discriminator in pix2pix. To reduce the overall parameter amount of the network, we adjust the initial output channel number of G*_s_* to 4; that is, the number of all the feature map channels in the network is 1/16 of the original U-net. In order to improve the calculation efficiency of the overall algorithm, this paper generally uses lightweight convolutions in the network, such as GConv, DSConv, BSConv, Ghost module, etc. In V2T-GAN, G_1_ has the largest overall network parameters, and its lightweight operation has the greatest impact on the overall network. Therefore, we analyzed and compared the different lightweight methods of G_1_, and finally adopted GConv, and the standard convolution of G_1_, G*_s_* and D_1_ are all replaced by GConv with a group number of 4.

There have been many research studies on lightweight convolutions, and the methods applied in this paper will be introduced below.

The specific implementation of GConv is divided into three steps:GConv divides the input channels into even and non-overlapping groups according to the grouping number *g*;Perform standard convolution independently on each group that has been divided;Concat the results of the standard convolution in the dimension of the channel.

Depthwise Convolution (DConv) [32] is a special type of GConv. The number of groups and output channels are the same as the number of input channels. The DSConv consists of two steps:The first step is to perform DConv;Use standard convolution with a 1 × 1 convolution kernel to adjust the number of output channels.

The BSConv is also divided into two steps:
First perform standard convolution with a 1 × 1 convolution kernel to adjust the number of output channels;Use DConv.

The GhostModule is divided into three steps:
Perform standard convolution with a 1 × 1 convolution kernel. The number of output channels in this step is: C_1_ = [C*_in_*/*r*], where C_1_ is the number of output channels in the first step, C*_in_* is the number of input channels, and *r* represents the manually set rate;Use GConv on the output result of the first step, the number of groups is the number of output channels in the first step (that is, equal to C_1_), the number of output channels in this step is: C_1_ × (*r* − 1);Concatenate the output result of the first step and the second step to get the final result.

### 3.2. Second-Level Network

The second-level network further optimizes the infrared image output by the first-level network, and uses the GIV images as an auxiliary task to guide the second-level target task generative network to predict infrared images with more accurate details and textures. The network structure is shown in the purple part of Figure 1, including a target task generator G_2_, a discriminator D_2_ and an auxiliary task network G*_g_*. The input of G_2_ is concatenated with the predicted infrared image T_1_ of the first-level network, the output feature map of the penultimate layer of G_1_ and the GIV image I*_g_*, and finally the predicted infrared image T_2_ is output. D_2_ has the same structure and function as D_1_, which used to discriminate the predicted infrared image T_2_ and the target infrared image T*_true_*. G*_g_* estimates the GIV image from T_2_, and guides G_2_ to further optimize the predicted infrared image.

In the case of good lighting, visible light images have more detailed texture information than infrared images. GIV images also have rich detailed texture information, and compared with visible light images, it has a high similarity with infrared images in terms of human visual effects and some objective metrics, such as FID [33], LIPIS [34], etc. Therefore, we adopt the GIV image as the auxiliary task of the second-level network to guide G_2_ to further optimize the detailed texture information of the predicted infrared image.

#### 3.2.1. Target Task Generator G_2_

An illustration of the second-level target task generator G_2_ is depicted in Figure 2, consisting of an MFM module [28], four L-FMR modules that add skip connection and a Ghost module. The MFM module is shown in Figure 3. In order to obtain the information of multiple receptive fields, the input is respectively passed through four dilated convolutions with a convolution kernel size of 3 × 3 and a dilation rate of 1, 2, 3 and 4. The output of the dilated convolution with different dilation rates are added and fused, and then the added results are concatenated. The input and output of G_2_ are similar. To increase the direct mapping between input and output, the input is added to the final concatenated result after a pointwise convolution. The L-FMR module is improved from the FMRB [35], which is a network module for image deblurring tasks. It has been verified that FRMB can learn and restore the detailed texture information of the image. In order to reduce the amount of overall network parameters, we replace all the standard convolutions in FMRB with a Ghost module with a rate of 4, which is the L-FMR module in Figure 2.

#### 3.2.2. Second-Level Auxiliary Task Network G*_g_*

In order to better guide G_2_ to learn the detailed texture information and obtain the predicted infrared image T_2_ with rich detailed texture, G*_g_* only needs to pay attention to the details of T_2_. Therefore, the receptive field of G*_g_* should be smaller. The network structure of G*_g_* is shown in Figure 4. The upper part is the overall network structure of G*_g_*, which contains four blocks, and the number in the middle represents the number of channels. The lower part represents the specific network structure of each block: it contains three cascaded Ghost modules, the number is the size of the convolution kernel and the convolution step length is 1. This kind of network structure makes the overall network receptive field size of G*_g_* only 5 × 5, and the parameter quantity is extremely small.

### 3.3. Thrid-Level Network

To further optimize the predicted infrared image and obtain a clear edge, the third-level network adds the edge image of visible light as auxiliary information. At the same time, inspired by [36], we learn the position offset information to further obtain infrared images with sharper and more accurate edges. As shown in the green part of Figure 1, the third level has just one displacement network, G*_d_*. The input of G*_d_* is concatenated with the predicted infrared image T_2_ of the second level network and the edge image of the visible light image. The output is the positional offset map of the input image in the row direction and the column direction. Then, the input image T_2_ is resampled from the two position offset maps to obtain the final predicted infrared image T_3_.

The overall network structure of G*_d_* is shown in Figure 5a, using a codec network structure, and the encoding end includes four down-sampling residual blocks (D-Res). The specific network structure of D-Res is shown in Figure 5b. The input goes through two standard convolutions with 3 × 3 convolution kernels, and then through a bilinear interpolation down-sampling to compress the resolution of the feature map twice. Finally, the skip connection of the convolution with a convolution kernel of 4 × 4 and step size of 2 is added to the down-sampling result. The network structure of the decoding end is symmetrical to the encoding end, including four up-sampling residual blocks (U-Res). The specific network structure of U-Res is shown in Figure 5c. The input goes through two standard convolutions with 3 × 3 convolution kernels, and then through a bilinear interpolation up-sampling to double the resolution of the feature map. Finally, the deconvolution skip connection with a convolution kernel of 4 × 4 and step size of 2 is added to the up-sampling result.

In this paper, according to the row direction position offset map, I*_Row_*, and the column direction position offset map, I*_Col_*, predicted by G*_d_*, the second-level output result T_2_ is resampled to obtain the final predicted infrared image, T_3_. The resampling process is defined as Equation (1). T_3_ (*x*, *y*) represents the gray value of the third-level network output image at the position (*x*, *y*); and T_2_ (*x*, *y*) represents the gray value of the second-level network output image at position (*x*, *y*). Row (*x*, *y*) and Col (*x*, *y*) denote the position offset in the row and column direction.
T_3_(*x*, *y*) = T_2_(*x* + Row(*x*, *y*), *y* + Col(*x*, *y*)),(1)

### 3.4. Loss Function

The three-level network in this paper is jointly trained in an end-to-end manner. The gradient descent of the discriminator and the generator is performed alternately; we first fix the parameters of D_1_ and D_2_, train G_1_, G*_s_*, G_2*,*_ G*_g_* and G*_d_*, and then fix G_1_, G*_s_*, G_2_, G*_g_* and G*_d_*, and train D_1_ and D_2_. The overall loss function L*_final_* uses a minimum–maximum training strategy, and the expression is as follows:(2)$$\underset{\left\{G_{1},G_{s},G_{2},G_{g}\right\}}{\min}\underset{\left\{D_{1},D_{2}\right\}}{\max}L_{final}=L_{GAN}+L_{pixel},$$
L*_GAN_* is the sum of adversarial loss functions, and L*_pixel_* is the sum of pixel-level loss functions. L*_GAN_* includes the first-level adversarial loss, L*_GAN_*_1_, and the second-level adversarial loss, L*_GAN_*_2_. The expression is as follows:(3)$$L_{GAN}=L_{GAN1}+10\times L_{GAN2}.$$

The first-level discriminator D_1_ is used to distinguish the synthetic image pair [I*_rgb_*, T_1_] and the real image pair [I*_rgb_*, T*_true_*]. The loss function adopts the combination of cross entropy, which is expressed as
(4)$$\begin{array}{l} L_{GAN1}=E_{I_{rgb},T_{true}}\left[\mathrm{logD}(I_{rgb},T_{true})\right]\\ \;\;\;\;\;\;\;\;\;\;\;+E_{I_{rgb},T_{1}}\left[\log(1-D(I_{rgb},T_{1}))\right] \end{array}$$

The second-level discriminator D_2_ is used to distinguish the synthetic image pair [I*_rgb_*, T_2_] and the real image pair [I*_rgb_*, T*_true_*], expressed as
(5)$$\begin{array}{l} L_{GAN2}=E_{I_{rgb},T_{true}}\left[\mathrm{logD}(I_{rgb},T_{true})\right]\\ \;\;\;\;\;\;\;\;\;\;\;+E_{I_{rgb},T_{2}}\left[\log(1-D(I_{rgb},T_{2}))\right] \end{array}$$

The total pixel-level loss function L*_pixel_* includes the L_1_ loss function L_G__1_ and L_G*s*_ of the first-level generative network G_1_ and Gs; the L_1_ loss function, L_G__2_ and LG*_g_*, of the second-level generative network, G_2_ and G*_g_*; the gradient loss function L*_g_*__G2_, which is more sensitive to texture; and the L_1_ loss function L_G*d*_ after resampling. The expression is defined as follows:(6)$$\begin{array}{l} L_{pixel}=\lambda_{1}L_{G1}+\lambda_{2}L_{Gs}+\lambda_{3}L_{G2}\\ \;\;\;\;\;\;\;\;\;+\lambda_{4}L_{Gg}+\lambda_{5}L_{g\_G2}+\lambda_{6}L_{Gd} \end{array}$$
λ is a hyperparameter, which represents the weight of each loss function. G_1_, G_2_ and G*_d_* are the target task networks with the highest weights; the networks G*_s_* and G*_g_* are responsible for auxiliary tasks and have lower weights; the gradient loss function is used to increase the network’s ability to perceive edges, with the smallest weights. After experiments, we finally set λ from 1 to 6 as 100, 5, 200, 10, 0.5 and 100, respectively. The L_1_ loss function represents the average absolute error, expressed as
(7)$$L_{1}=\frac{1}{N}\sum_{i=1}^{N}\left|y_{i}-y_{i}^{*}\right|,$$
where *i* is the pixel index, *N* is the total number of all pixels in an image, and $y_{i}$ and $y_{i}^{*}$, respectively, represent the real and predicted gray value at pixel *i*. The expression of the gradient loss function L*_g_*__G2_ is as follows:(8)$$L_{g\_G2}=\frac{1}{2N}\sum_{i=1}^{2N}\left(\left|\nabla_{h}y_{i}-\nabla_{h}{\hat{y}}_{i}\right|+\left|\nabla_{v}y_{i}-\nabla_{v}{\hat{y}}_{i}\right|\right),$$

$\nabla_{h}{\hat{y}}_{i}$ and $\nabla_{h}y_{i}$ represent the gradient value in the horizontal direction at pixel *i* of the target infrared image T*_true_* and the infrared simulation image respectively.

## 4. Experiments

### 4.1. Experimental Details and Evaluation Metrics

#### 4.1.1. Dataset

We performed the experiments on MPD [11], which consists of image pairs for visible light images and corresponding infrared images with a resolution of 640 × 512. The training set and test set contain 50,187 and 45,141 image pairs, respectively. Both the training set and the test set involve three scenes—campus, street and suburbs—and each scene contains images taken during the day and night. We select image pairs in the daytime as the training set of the network, and the training set size consisted of 33,399 image pairs. Correspondingly, we randomly select 565 image pairs from the daytime image pairs in the MPD test set as the test set of the network. We resize the image resolution to 256 × 256 through bilinear interpolation down-sampling.

Predicting the semantic segmentation image and gray-scale inversion image of visible light is the auxiliary task of this network. The gray-scale inversion image is obtained by converting the visible light image from a color image to a gray-scale image and then performing the gray-scale value inversion operation. Semantic segmentation images can be predicted by feeding visible light images into a model trained by Refinenet [37] on Cityscapes. Cityscapes is a large dataset mainly used for semantic segmentation. The main scene is outdoor streets, similar to MPD. The edge image of the visible light is the auxiliary information in the third-level network, which is extracted by the Canny operator with the upper and lower thresholds set to 60 and 120, respectively.

#### 4.1.2. Evaluation Metrics

In the previous work of the image domain conversion task, there are some recognized evaluation metrics to evaluate the similarity between the network predicted image and the real target image. We used the mean absolute relative error (*Rel*), mean log10 error (*Log*10), root mean squared error (*Rms*) and accuracy index (*δ* < 1.25*^i^*, *i* = 1, 2, 3). The calculation expressions of each metrics are as follows:(9)$$Rel=\frac{1}{\left|N\right|}\sum_{i=1}^{N}\left|y_{i}-y_{i}^{*}\right|/y_{i}^{*},$$
(10)$$Log10=\frac{1}{\left|N\right|}\sum_{i=1}^{N}\left|\mathrm{lg}y_{i}-\mathrm{lg}y_{i}^{*}\right|,$$
(11)$$Rms=\sqrt{\frac{1}{\left|N\right|}\sum_{i=1}^{N}{\left\|y_{i}-y_{i}^{*}\right\|}^{2}},$$
(12)$$\delta =max(\frac{y_{i}}{y_{i}^{*}},\frac{y_{i}^{*}}{y_{i}})<thr,$$
where *i* is the pixel index, and *N* is the total number of pixels in an infrared image. $y_{i}$ and $y_{i}^{*}$ respectively, represent the gray value of the target image and the gray value of the predicted image at pixel *i*. We also employed pixel-level similarity metrics to evaluate our method, i.e., Structural-Similarity (SSIM) and Peak Signal-to-Noise Ratio (PSNR). PSNR and SSIM, as evaluation metrics for image deblurring and super-resolution, can better reflect the similarity of the two images.

#### 4.1.3. Training Setup

Our method was implemented with Pytorch using one NVIDIA GeForce RTX 2080 Ti GPU with 16 GB memory. We used a Gaussian distribution with a mean of 0 and a standard deviation of 0.2 for weight initialization. We minimized the loss function using the Adam optimizer with a momentum of 0.5 and initial learning rate of 0.0001. We set the batch size to 4.

### 4.2. Results

This section compares our method with other state-of-the-art image domain conversion methods based on generative adversarial networks. The comparison results are shown in Table 1. Pix2pix [9] is a popular cGAN that can realize image-to-image conversion and is suitable for all image domain conversion tasks. The input of the network is conditional information. In this paper, the input of pix2pix is set as a visible light image, and the output is a corresponding infrared image, without other auxiliary tasks or auxiliary information. X-Fork [38] is a GAN that realizes cross-view image translation and requires auxiliary tasks of semantic segmentation. Selection-GAN [24] is also a GAN that realizes cross-view image translation, and its network structure is a two-level GAN, where each level of the network is guided by an auxiliary task of semantic segmentation. SEAN [39] can achieve image fusion and conversion. The style image needs to be added as auxiliary information in the process of converting the input image to the target image. In this paper, the semantic segmentation image is used as input, the GIV image is used as the style image, and the output is the predicted infrared image. LG-GAN [40] explores the generation of scenes in the local environment, and considers the global and local context at the same time, which can effectively deal with the generation of small objects and scene details.

As can be seen from Table 1, compared with other advanced generative adversarial networks for image domain transformation, our algorithm achieves the best results on various objective evaluation metrics.

The visual comparison between our method and other advanced algorithms is shown in Figure 6. Our proposed V2T-GAN has the smallest network parameters, only 15.24 M, and the lowest error RMS. The overall parameters of the network are about 73.35%, 73.68%, 73.83%, 94.29% and 14.03% lower than the Pix2pix, X-Fork, Selection-GAN, SEAN and LG-GAN algorithms, respectively.

### 4.3. Ablation Study

To further analyze the details of the proposed approach, ablation experiments were conducted by investigating different configurations of the components of V2T-Net.

#### 4.3.1. Three-Level Network Structure

To verify the effectiveness of the three-level network structure, this section compares the experimental results of the first-level network, the two-level network and the third-level network. The comparison results are show in Table 2. We can observe the improvement in the three-level network structure in this table, which outperforms other structures in all the metrics.

The predicted infrared images of the one-level and two-level networks are shown in Figure 7. It can be seen that the results of the one-level network are relatively rough, while the results of the two-level network are more accurate in detail and more similar to the target image. For example, for the road signs selected in the first image, part of the structure is missing in the result of the first-level network, and the outline of the two-level network is more complete. The framed parts in the second image include road signs and branches. Comparing the two results, we can observe that the detailed texture information of the two-level network results is relatively more accurate.

Figure 8 shows the infrared simulation images output by the two-level network and three-level network. We can see that the visualization results of the selected area in the yellow box after the position offset optimization are poor, mainly reflected in the blurred image edges, unclear textures and many errors at the edges. This is because the position offset network adopts the CNN training method; that is, it learns to convert the image directly through the pixel-level loss function. Although the converted result performs better on the pixel-level objective indicators, the subjective perception of the human eye is poor. From the perspective of the local image, it can be found from the cars selected in the second and fourth columns that the contour of the two-level network conversion result is easier to identify and more similar to the target infrared image.

#### 4.3.2. Auxiliary Task

Auxiliary tasks are added to the method in this paper to improve network performance. In this section, we compare the effects of auxiliary tasks. The auxiliary task of the first-level network is semantic segmentation of images, and the auxiliary task of the second-level network is the GIV images. The results of specific ablation experiments are shown in Table 3: Structure 1, removing the two auxiliary tasks of semantic segmentation and GIV images at the same time; that is, removing the G*_s_* and G*_g_* networks. Structure 2, only remove the GIV image, which means there is no G*_g_* network. Structure 3, only remove the semantic segmentation image; that is, no G*_s_* network. Row 4 represents the complete V2T-GAN.

It can be seen from Table 3 that the our complete V2T-GAN, including semantic segmentation and GIV image auxiliary tasks, obtains the best experimental results. The accuracy rate *δ* < 1.25 is 2.30%, 1.96% and 1.30% higher than Structures 1–3, respectively. The structure one has no auxiliary tasks, and the performance is the worst. Structure 3 is better than the Structure 2 network in various objective metrics, indicating the GIV auxiliary task has a greater effect than semantic segmentation.

Although the auxiliary task of semantic segmentation in Structure 2 enables the network to learn more correct structure information, the calculation process of objective metrics cannot add weight to the structure information. The auxiliary task of GIV image in Structure 3 enables the network to obtain more detailed image information. Even if there are some differences in structure, it can still ensure better metrics. This is also a limitation of objective metrics.

#### 4.3.3. Edge Auxiliary Information

In order to guide the third-level network to learn more clear and accurate edge information, the input of the third-level network adds the edge image of visible light as auxiliary information. We conducted an experimental analysis on the effectiveness of the edge image auxiliary information, and the results are shown in Table 4. We found that adding visible light edge images as auxiliary information can improve the objective metrics of predicting infrared images, which means that V2T-GAN has indeed learned a sharp edge from this auxiliary task.

#### 4.3.4. Lightweight Convolution

To reduce the amount of overall network parameters, we generally use lightweight convolution in the proposed network. The two sub-networks with the largest amounts of parameters in V2T-GAN are G_1_ and G*_s_*. Therefore, this section compares the different lighweight methods of G_1_ and G*_s_*. The experimental results are shown in Table 5. BSConv, DSConv, GhostModule and GConv, respectively, represent the replacement of the standard convolutions in G_1_ and G*_s_* with blueprint separable convolution, depthwise separable convolution, GhostModule and group convolution. In the experiment, the grouping number of group convolution and GhostModule were both set to 4.

It can be found from Table 5 that the overall network parameters using BSConv are the smallest, but the overall network using GConv performs best in various objective metrics. The goal of a lightweight network for V2T-GAN is to reduce the amount of overall network parameters, improve calculation efficiency, alleviate the problem of network overfitting and improve conversion accuracy. In order to trade off accuracy and efficiency, we finally use GConv in our network.

## 5. Conclusions

We propose a three-level refined lightweight GAN with cascaded guidance (V2T-GAN) to address image domain conversion task on a visible light image to the corresponding infrared simulation image. In the three-level network, semantic segmentation images, GIV images and visible light edge images were used as input information for auxiliary tasks. The experimental results on the MPD show that our method obtains much better results than the state-of-the-art on the task of feature conversion from visible light to infrared images.

In the future, we would like to be able to convert from visible light to infrared images without having to create a one-to-one mapping between the training data, as well as apply the idea of the algorithm in this paper to other fields, such as the fusion of visible light and infrared images and the detailed enhancement of infrared images.

## Figures and Tables

**Figure 1 sensors-22-02119-f001:**
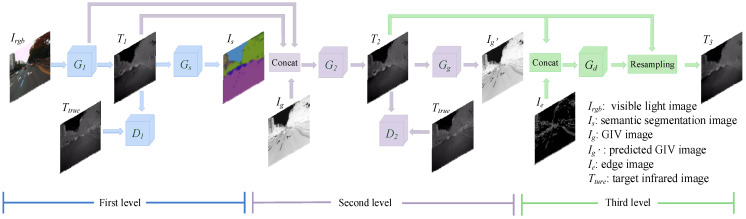
V2T-GAN network structure.

**Figure 2 sensors-22-02119-f002:**
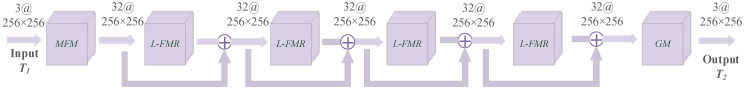
The proposed network of G_2_.

**Figure 3 sensors-22-02119-f003:**
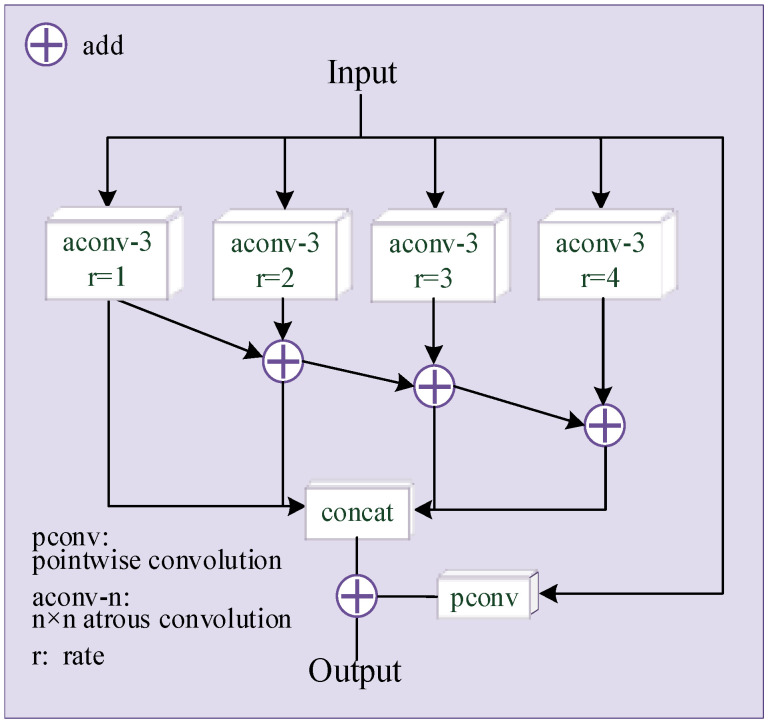
The structure of MFM.

**Figure 4 sensors-22-02119-f004:**
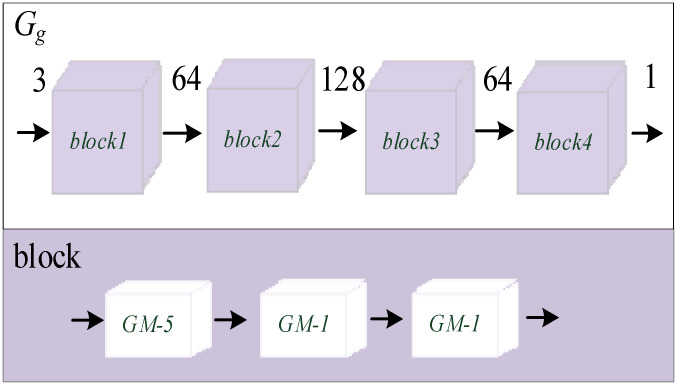
The proposed network of G*_g_*.

**Figure 5 sensors-22-02119-f005:**
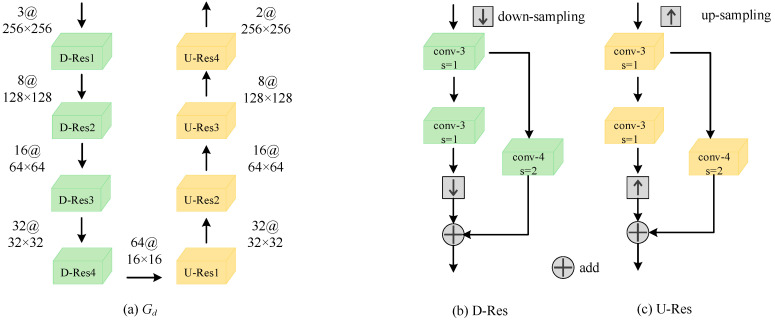
The proposed network of G*_d_*.

**Figure 6 sensors-22-02119-f006:**
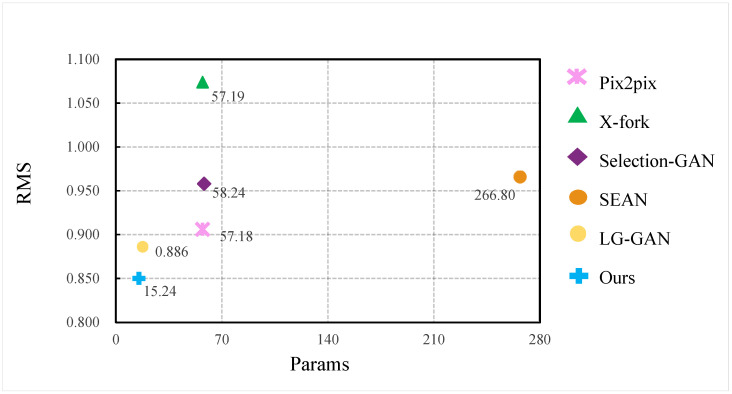
Compared with the computational efficiency of advanced algorithms.

**Figure 7 sensors-22-02119-f007:**
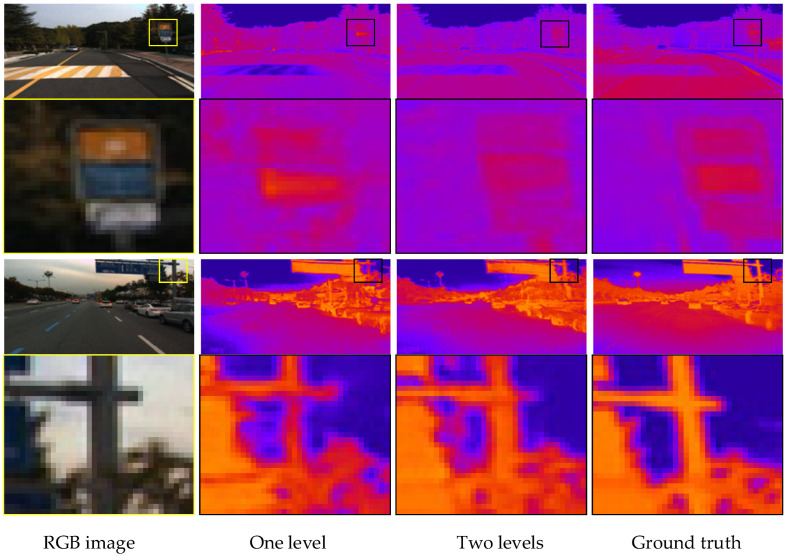
Comparison of the one-level and two-level networks’ visualization results.

**Figure 8 sensors-22-02119-f008:**
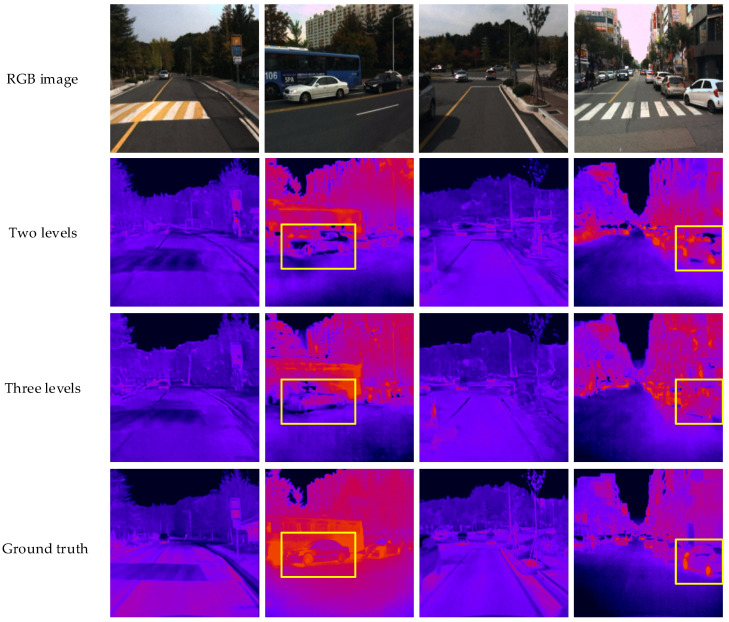
Comparison of two-level and three-level networks’ visualization results.

**Table 1 sensors-22-02119-t001:** Comparison of the algorithms in objective metrics.

Methods	The Lower, The Better			The Higher, The Better		
	Abs Rel	Avg log10	RMS	*δ* < 1.25	PSNR	SSIM
Pix2pix [9]	0.248	0.107	0.906	0.571	22.431	0.985
X-Fork [38]	0.314	0.130	1.074	0.480	20.692	0.984
Selection-GAN [24]	0.284	0.112	0.958	0.554	21.976	0.982
SEAN [39]	0.293	0.114	0.966	0.564	21.804	0.983
LG-GAN [40]	0.262	0.102	0.886	0.616	22.601	0.989
Ours	0.247	0.099	0.850	0.623	22.908	0.990

**Table 2 sensors-22-02119-t002:** Comparison of the different network structures.

Network Structure	The Lower, The Better			The Higher, The Better		
	Rel	Avg log10	RMS	*δ* < 1.25	PSNR	SSIM
One-level	0.254	0.100	0.859	0.617	22.838	0.988
Two-level	0.254	0.099	0.853	0.619	22.872	0.990
Three-level	0.247	0.099	0.850	0.623	22.908	0.990

**Table 3 sensors-22-02119-t003:** Comparison of the different auxiliary tasks.

Setup	The Lower, The Better			The Higher, The Better		
	Rel	Avg log10	RMS	*δ* < 1.25	PSNR	SSIM
−G*_s_*, G*_g_*	0.257	0.103	0.876	0.609	22.674	0.989
−G*_g_*	0.255	0.102	0.870	0.611	22.678	0.989
−G*_s_*	0.249	0.101	0.855	0.615	22.811	0.990
Ours	0.247	0.099	0.850	0.623	22.908	0.990

**Table 4 sensors-22-02119-t004:** Effectiveness of the edge auxiliary information.

Methods	The Lower, The Better			The Higher, The Better		
	Abs Rel	Avg log10	RMS	*δ* < 1.25	PSNR	SSIM
−*I_e_*	0.248	0.099	0.850	0.616	22.922	0.989
Ours	0.247	0.099	0.850	0.623	22.908	0.990

**Table 5 sensors-22-02119-t005:** Comparison of the different lightweight convolutions.

Methods	The Lower, The Better			The Higher, The Better			Params
	Abs Rel	Avg log10	RMS	*δ* < 1.25	PSNR	SSIM	
BSConv	0.256	0.105	0.898	0.601	22.442	0.989	5.081M
DSConv	0.259	0.108	0.916	0.593	22.257	0.989	5.126M
GhostModule	0.260	0.105	0.891	0.604	22.565	0.987	15.263M
GConv	0.247	0.099	0.850	0.623	22.908	0.990	15.235M

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
