# Peer review of "V2T-GAN: Three-Level Refined Light-Weight GAN with Cascaded Guidance for Visible-to-Thermal Translation"

_sensors, 2022, doi:10.3390/s22062119_

Round 1

Reviewer 1 Report

A very actual, interesting and important research subject, presenting in the paper a practical solution. Wrking on that subject is pretty hard so, all the research is appreciated. Though the abbreviation GAN should be first of all explained, because not all the eventual readers work on that subject.

The paer is well documented, references properly used

In 3.1 line 170 authors say “in THREE steps” – please explain better, because below that line, moreideas are presented, so it’s not clear at all what the THREE steps are. So for all the structure lines 170-192 an indented structure should be chosen, so that the specific steps for all procedures should be visible

In line 310 please explain MPD =?

All experiments and results clearly and suggestively presented.

Reviewer 2 Report

In this manuscript, the Authors propose a deep learning approach to simulate thermal images from optical images (visible-to-thermal translation). 

The paper is well written although a strong revision to correct minor issues is required (see below). The introduction is fine, well addressing related problems and the statement of the problem seems correct.

Results are also fine, on many images, using objective metrics and comparing to other closely related techniques.

Although results are obvious, this reviewer does not understand how a thermal image can be obtained from an optical image without knowing relevant information that seems mandatory: temperature. This reviewer would like to see a result for an optical image taken at different conditions (temperature, illuminance....) and know the ground truth and then, get the result from the proposed method and extract conclusions. Otherwise, this kind of ill-posed problem seems useless.

Some authors appeared self-cited 4 times; is that strictly necessary?

Some format issues and suggestions:

- line 68, "correct structure; The..:",--> correct,
- line 75, "seg‐ 76 ment‐ed; ..:", line 86, "be‐tween.."
There are many more: correct them all,

- line 141, "T2 in the x and y directions.."; write x and y in italic,

References:
- some format issues require attention (ref. 12, 13,...).

Reviewer 3 Report

This paper proposed “V2T‐GAN: Three‐level Refined Light‐weight GAN with Cascaded Guidance for Visible‐to‐Thermal Translation”. The proposed idea is interesting. The following questions should be taken into consideration.

  1. The abstract and introduction to the topic of the work is written well, however the abstract section also be included the some details/name of dataset used for this research.
  2. Introduction section should also include the organization of other parts of article at the end of introduction section.
  3. As author proposed “Three‐level Refined Light‐weight GAN with Cascaded Guidance ” for Visible‐to‐Thermal Translation generally so much complex approach consists of many parameters like performance and Robustness, have you make sure these factors inclusion in your work, please clarify?
  4. Moreover how the author combine the diversity of different models for outcomes?
  5. Figure 1 is not clear please make it more visible with details?
  6. Did author used any preprocessing technique for dataset preprocessing, explain please?
  7. As the dataset images size is 640×512, did author used any specific machine to reduce the sizes of the images?
  8. Please define the units of the values in Table 1.
  9. Is dataset publically available for other user can be used to verify the results?
  10. Conclusion should also include future direction.
  11. Proof read, remove typo errors and spelling mistakes

Round 2

Reviewer 2 Report

All concerns were attended.

Reviewer 3 Report

Dear Authors,

You have done revision well, please make sure the article needs minor language proofread.